# Flood Forecasting and Warning System Structures: Procedure and Application to a Small Urban Stream in South Korea

**Yangho Song [1]**, **Yoonkyung Park [2]**, **Jungho Lee [1]**, **Moojong Park [3]** and **Youngseok Song [4],***

[1]  Department of Civil and Environmental Engineering, Hanbat National University, Daejeon 34158, Korea
[2]  Corporate-Affiliated Research Institute, Deepcloud Co., Ltd., Busan 48058, Korea
[3]  Department of Aeronautics and Civil Engineering, Hanseo University, Seosan 31962, Korea
[4]  Department of Civil Engineering and Landscape Architectural, Daegu Technical University, Daegu 42734, Korea
*   Correspondence: kind711@hanmail.net; Tel.: +82-53-560-3876

**Abstract:** The runoff from heavy rainfall reaches urban streams quickly, causing them to rise rapidly. It is therefore of great importance to provide sufficient lead time for evacuation planning and decision making. An efficient flood forecasting and warning method is crucial for ensuring adequate lead time. With this objective, this paper proposes an analysis method for a flood forecasting and warning system, and establishes the criteria for issuing urban-stream flash flood warnings based on the amount of rainfall to allow sufficient lead time. The proposed methodology is a nonstructural approach to flood prediction and risk reduction. It considers water level fluctuations during a rainfall event and estimates the upstream (alert point) and downstream (confluence) water levels for water level analysis based on the rainfall intensity and duration. We also investigate the rainfall/runoff and flow rate/water level relationships using the Hydrologic Engineering Center's Hydrologic Modeling System (HEC-HMS) and the HEC's River Analysis System (HEC-RAS) models, respectively, and estimate the rainfall threshold for issuing flash flood warnings depending on the backwater state based on actual watershed conditions. We present a methodology for issuing flash flood warnings at a critical point by considering the effects of fluctuations in various backwater conditions in real time, which will provide practical support for decision making by disaster protection workers. The results are compared with real-time water level observations of the Dorim Stream. Finally, we verify the validity of the flash flood warning criteria by comparing the predicted values with the observed values and performing validity analysis.

**Keywords:** flash flood forecasting; flood forecasting and warning system; flood warning criteria; rainfall/runoff

## 1. Introduction

The extent of flood damage caused by locally concentrated heavy rainfall has been increasing owing to extreme weather events such as heavy rainfall and snowfall, which have become more frequent because of climate change. The risk of flood damage is also rising because of increased runoff due to continuing urbanization and limited pervious spaces for absorbing and holding stormwater. Flood damage reduction measures for urban areas can be categorized into structural and non-structural methods. Typical structural measures include the maintenance of streams and embankments and maintenance and extension of drainage facilities including pipelines.

Non-structural measures are means of actively preparing for and responding to anticipated weather events by methods such as decision-making systems, for example, flash flood forecasting and warning systems [1].

Given that most riverside areas in urban watersheds are used as roads or residential areas, it is easier to apply non-structural measures such as flash flood forecasting than short-term structural measures. In particular, Korea is characterized by great differences between its maximum and minimum precipitation amounts, making it prone to floods and droughts. The precipitation in summer (June to September) accounts for approximately 70% of the annual precipitation and heavy rainfall events during this period cause increasingly frequent flood/inundation damage. It is essential to predict such events accurately and to provide detailed forecasting. That is, a decision support system based on accurate prediction and forecasting is necessary to minimize the damage caused by locally concentrated and sporadic heavy rainfall events.

Most previous researches have focused on investigating the causes of floods using rainfall-runoff models. In some studies, physical models have been developed for flood forecasting by considering regional characteristics or specific parameters. These investigations involved rainfall-runoff simulations taking into account climate change [2,3]. Attempts have been made to provide results with high reliability for decision making by accurately predicting the changes in runoff intensity and frequency [4,5]. Zahmatkesh et al. [6] proposed a rainfall-runoff model based on K-means clustering to minimize the uncertainties of the parameters constituting the model and applied it to the Bronx River watershed, New York City. Using this model, the runoff based on the predicted rainfall was simulated to predict the future runoff for the watershed. The runoff simulation using the clustering scheme was found to be up to 50% more accurate than the simulations performed using individual models. Zhang and Yang [7] used genetic algorithms to derive a unified value from eight different rainfall-runoff models. They applied the multi-model ensemble method to the Yellow River Basin and demonstrated that it outperformed individual rainfall-runoff models in terms of prediction reliability. Mehr and Kahya [8] and Mehr and Nourani [9] also proposed hybrid models using genetic algorithms on rainfall-runoff modeling results. In all of these studies, the extents of the parameter uncertainties were assessed at each stage of the rainfall-runoff analysis and their impacts on the runoff analysis were evaluated.

The current research trend is to integrate new techniques into rainfall-runoff models. For example, artificial neural network (ANN) modeling has been used for flood forecasting based on the water levels of rivers/streams [10,11]. Wu et al. [12] presented an ANN-based method to predict the variations in watershed runoff and stream water level and proposed a regression equation for water level prediction. Aichouri et al. [13] compared various ANN-based models for rainfall-runoff analysis. They pointed out that the models compared in the study mostly dealt with linear (regular) rainfall-runoff patterns and highlighted the need to consider nonlinear (irregular) rainfall-runoff. They additionally proposed an ANN-based model designed to overcome this limitation and listed the improved features of the developed model for irregular rainfall-runoff prediction based on the results obtained by applying the model to catchments in the Mediterranean region. Zhang et al. [14] used an Elman neural network to improve the prediction accuracy of the time-series runoff input data by taking into account the nonlinear characteristics of the data. They proposed a water level prediction model for warning purposes by considering the hidden layer for lead time (t − 1) along with the originally predicted time (t). Studies on ANN-based water level prediction focus on improving the reliability of the predicted values based on the input data (input layer) selection. Stated differently, there is a lack of research on the relative structural suitability of the input, hidden, and output layers and a comprehensive model based on an optimal structure is required. Therefore, flash flood forecasting based on simple prediction using ANN results alone has room for improvement.

With the improved computational capabilities resulting from advanced computer technology, studies on hybrid models combining geographic information system (GIS) technology and distributed

runoff models using spatially distributed data have recently received increasing interest [15–17]. Liu et al. [18] focused on runoff behavior (responses) during heavy rainfall events.

They noted that runoff behavior plays an important role in runoff routing and presented the results of diffusive transport using a GIS-based model by applying a grid mesh to the Attert catchment in Luxembourg. For model verification, they compared the results of runoff routing from each cell with the data collected over the previous 30 months, which yielded similar hydrographs for the predicted and observed values. Khalfallah and Saidi [19] predicted runoff probabilities according to GIS-based rainfall frequency mapping and presented the flood simulation results obtained by applying the prediction results to the Hydrologic Engineering Center's River Analysis System (HEC-RAS). Cohen et al. [20] regarded the longitudinal river slope as a key parameter that determines flood discharge variability. They proposed a methodology for calculating the river slope using the Global River-Slope dataset and used the modified river slope for modeling. They also presented regression analysis results depicting the correlations between the catchment parameters and river slope, which are necessary for flood discharge estimation. Prediction using a distributed model is very important because it takes into account the non-uniformity of the geomorphological spatial distribution. However, this prediction process is time consuming because its rainfall-runoff simulation requires hydrodynamic numerical analysis. Furthermore, studies on flood forecasting and warning systems using distributed models are limited to small sub-watershed areas or low resolution.

The UK and US provide user-centered customized flood forecasting based on flood warning, preparedness, and response systems in an effort to reduce flood damage. The UK provides flood risk and flood zone maps covering all riverine areas across the country and operates a three-step flood warning system [21]. In the US, flood warnings are issued based on 75% of the threshold water level using the streamflow observations received from over 6800 stream gauges in the Automated Flood Warning System network distributed across the country [22].

As described above, flood risk in urban watersheds is a topic of extensive research, and numerous studies have been conducted on flood warning and response systems. However, it remains necessary to improve the knowledge and application of the prediction methods. Most importantly, the results obtained from various studies should support flood prediction with adequate lead time. Locally concentrated, sudden, excessive rainfall causes rapid watershed response, which makes city centers vulnerable to flood damage. Therefore, flash flood forecasting is very important for ensuring sufficient lead time for disaster prevention.

The primary goal of flash flood forecasting is to prevent disasters that cause loss of human life and property. This objective can be achieved only by rapid and accurate transmission of reliable flood forecasting and warning information. Speed is a very important aspect of flood forecasting and warning. An adequate lead time ensuring timely response directly impacts the application of the flood forecasting and warning system. Models that combine parametric optimization, distributed models, and hydro-geomorphological features are inherently restricted from providing sufficient lead time for flood forecasting and warning. Research methods based on rainfall-runoff analysis are conclusively considered to be the most suitable for ensuring sufficient lead time and are highly effective in terms of practical aspects based on speed and accuracy.

The main purpose of this study was to develop a flood forecasting and warning system that supports rapid decision making. Specifically, the objective was to provide reliable information necessary for flood forecasting and warning by estimating flood and water levels with high accuracy based on a one-dimensional flood analysis model. The high-water marks of a stream were used to develop a stable flash flood warning operation plan. High-water marks are customarily used independently to collect local rain observation records or establish criteria for situation assessment. However, when used independently, high-water marks cannot function as reliable measures for flood warning because they can only help in assessing the current situation or indicate disaster situations that occurred in the past. Furthermore, damage by heavy rainfall should be reduced by providing forecasts with sufficient

lead time to ensure appropriate preparedness and response by considering the spatial movement and temporal evolution of rainfall, which conventional water level and rain gauge networks cannot provide.

In summary, it is necessary to establish a rainfall and flood prediction system that considers the upstream and downstream relationships between the currently used water level points to ensure sufficient lead time. Considering the growing need to establish a flash flood forecasting and warning system based on a reliable prediction system, the objective of this study was to develop a flood forecasting and warning method capable of integrated consideration of the past and current states of the upstream and downstream points.

Most urban watersheds are significantly influenced by the backwater conditions of their mainstream channels. Therefore, the accuracy of flood forecasting and warning for urban streams can be ensured only through meticulous analysis of the relationships between rainfall and runoff and between flow rate and water level along with their specific manifestations. In this study, we prepared an analysis system for flood forecasting and warning, which can be directly applied to heavy rainfall events, by establishing a method for hydrodynamic analysis of urban streams exposed to such backwater effects and by creating a database for rainfall rates depending on rainfall duration. The objective of the model is to enable flood forecasting and warning based on the threshold water level at a specified warning point along the stream.

In this research, a flood-prone area located in the typical urban area of Seoul, Korea, which is at risk of flood damage, was selected as the study area. The aim was to support decision makers in operating flood forecasting and warning systems by analyzing the conditions according to the temporal distributions of various rainfall types based on rainfall predictions and allowing sufficient lead time for evacuation. The ultimate goal is to minimize the loss of human life and property in flood events.

## 2. Materials and Methods

### 2.1. Identification and Selection of Criteria for Issuing a Flash Flood Warning

In small and medium-sized urban streams, including urban watersheds, eco-stream restoration projects are underway to revive urban areas by creating cultural spaces, aside from conventional projects for enhancing disaster prevention and river management. An eco-stream serves as leisure space for urban residents and includes cultural spaces such as sports facilities and promenades created along both sides of the stream by designing and utilizing levees. However, such urban eco-streams are prone to intermittent overflows due to sudden rise in water level even during minor rainfall events because of their narrow widths and the relatively small elevation differences between the minimum streambed elevations and the levees. Consequently, these streams pose high risks of flood damage, including loss of life, due to the rapid rise in water level and overflow caused by rainfall. Therefore, efficient flash flood forecasting and warning techniques are necessary, along with meticulous response planning.

In this study, we set up two criteria for issuing urban stream flash flood warnings: First, the flood points on the levees were used to designate the criteria for caution issuance; and second, the design flood levels of the stream were employed to establish the criteria for evacuation, considering the inundation damage caused by overflows. That is, the threshold water levels for issuing flood warnings and evacuation orders were the water level reaching the levees used by residents for exercise or walking and the design flood level, respectively.

### 2.2. Rainfall Threshold Estimation for Issuing Flood Warnings Based on Water Level Observations

For hydraulic and hydrologic analysis of urban small and medium-sized stream watersheds, we established an analysis scheme by interactively integrating the two most commonly used models, the Hydrologic Engineering Center's Hydrologic Modeling System (HEC-HMS) and the HEC's River Analysis System (HEC-RAS). When the inflow estimated using HEC-HMS passed through the channel, the water level at each observation point was determined using HEC-RAS (Figure 1).

The aim of this research was to construct a flood forecasting and warning system. Swift rainfall-runoff simulations under several design frequencies were needed to realize this aim. HEC-HMS, which is a semi-distributed hydrological model, is easy to build and it can mimic the flooding performance in a watershed using simple functions. In addition, HEC-HMS is used in flooding analysis for flood disaster prevention in South Korea. Therefore, this study adopted HEC-HMS as the rainfall-runoff model. The backwater effect on the water level was taken into account by adjusting the known water surface (WS) elevation of HEC-RAS for each scenario [23,24]. Although simulations of situations similar to those in real life can be performed by modeling the water levels using the unsteady flow simulation feature of HEC-RAS, this process is very time consuming. Therefore, we simulated the effect of the upstream backwater, which depends on the downstream water level, in each scenario as a steady flow. Despite the higher number of simulation runs for various water levels when simulating steady flow, this method is simpler than unsteady flow simulation.

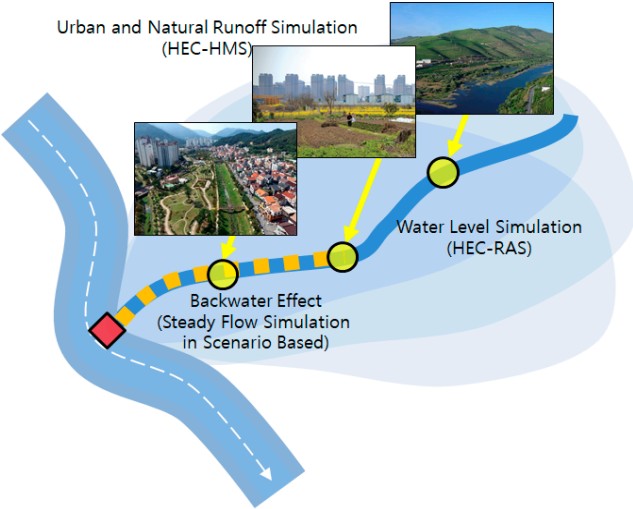

**Figure 1.** Hydraulic and hydrologic analysis using the Hydrologic Engineering Center's Hydrologic Modeling System (HEC-HMS) and the HEC's River Analysis System (HEC-RAS).

The flood forecasting and warning process is based on two water level points, as shown in Figure 2a: H1 is the warning point and H2 is the known WS at the downstream endpoint corresponding to H1, which is assumed to be a point at which real-time water level observations can be made. In Figure 2, the inflow rate (Q) at H1 depending on the backwater conditions (h2) at H2 is the preliminary data analysis point, given that the current inlet flow rate changes if h2′ at H2 changes under the same conditions as those of h1 at H1 (Q1 ≠ Q1′), as shown in Figure 2b. In such cases, the additional inflow rate (ΔQ′) and total flow rate (Q′) changes (Q ≠ Q′) with respect to the same water level rise (Δh) to reach the threshold water level for issuing a flood warning. Likewise, the rainfall threshold for issuing warnings changes (R ≠ R′). If the backwater effect increases due to the rise in water level downstream in the mainstream channel, the inflow rate at the upstream warning point decreases at the same water level. Consequently, the threshold water level can be reached even during a minor rainfall event.

When the analysis results are derived for each scenario by considering the inflow rate (Q) for each water level observed at H2 for each backwater condition using the HEC-HMS model, the additional inflow rate (ΔQ) according to the rise in water level (Δh) required to reach the threshold water level for flood warning is estimated by calculating the difference between the total flow rate (Q) and current inflow rate. Lastly, the rainfall conditions generating ΔQ are derived from the rainfall rate (R) for each rainfall duration pre-analyzed using the HEC-HMS model. In summary, the rainfall threshold for flash flood warning at a sub-watershed point can be estimated from the observations of the current water levels at the upstream and downstream high-water marks based on past analysis data. Encroaching water level predictions can be tabulated for each rainfall duration through hydraulic and hydrologic

analysis of the h2–h1–R relationships. These analysis steps can be employed to estimate the water level fluctuations during a flood event of the relevant stream and to issue an appropriate flood warning rapidly. Consequently, the short-term rainfall predictions obtained in this study can serve as flash flood warning issuance criteria if the real-time upstream and downstream water level observations correspond to the pre-tabulated rainfall rates plotted. The analysis procedure is summarized in Table 1 below.

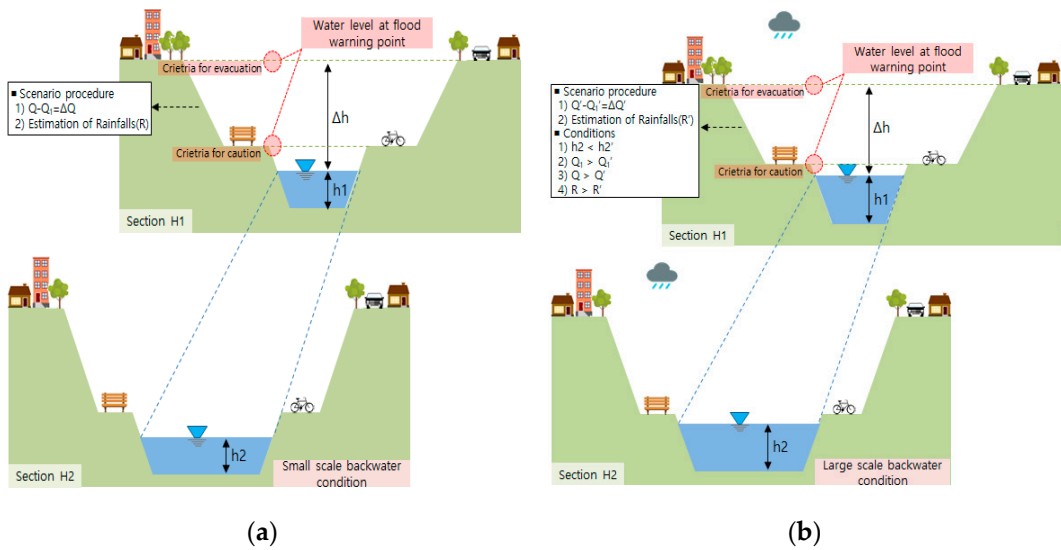

(**a**)　　　　　　　　　　　　　　　　　　　　　　(**b**)

**Figure 2.** Rainfall threshold analysis method for flood warning. (**a**) Analysis concept of rainfall threshold, (**b**) Rainfall threshold changed according to backwater conditions.

**Table 1.** Water level fluctuation analysis process based on upstream and downstream water level observations.

| Step | Analysis Process |
| --- | --- |
| STEP 1 | Real-time observation of water levels h1 and h2 at high-water marks H1 and H2, respectively |
| STEP 2 | Estimation of the inflow rate Q1 at H1 |
| STEP 3 | Estimation of the rise in water level Δh from h1 to the threshold water level for flood warning |
| STEP 4 | Estimation of the additional inflow rate ΔQ required for Δh |
| STEP 5 | Estimation of the rainfall rate R at each rainfall duration interval to reach ΔQ through direct runoff |
| STEP 6 | Flood warning issuance when the estimated rainfall rate is equal to the threshold rainfall rate R |

## 3. Watershed Characteristics and Application

This section describes how the rainfall threshold can be estimated based on the upstream and downstream high-water marks using relevant scenarios. To this end, it was necessary to employ real-time water level observation stations installed in the watershed considered in the study.

In addition to the water level observation stations operated for flood control, stations at high-water marks for water level observations, such as rainwater pumping stations operated by the local government, can also be used.

### 3.1. Watershed Characteristics

Dorim Stream is a tributary of a branch of the Anyang Stream. Its watershed covers 4250 ha, and the total channel length is 14.51 km. It flows through the major high-population-density areas of Seoul, covering five districts: Gwanak-gu, Guro-gu, Geumcheon-gu, Dongjak-gu, and Yeongdeungpo-gu. The Dorim Stream watershed has no sub-streams because it flows through the city center, where most small water bodies are covered. Figure 3 and Table 2 show the characteristics of the Dorim Stream in detail.

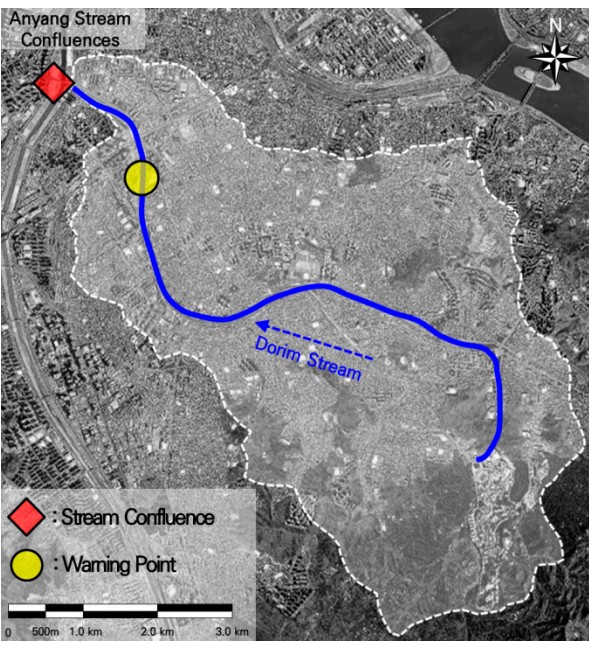

**Figure 3.** Watershed of Dorim Stream.

**Table 2.** Watershed properties of Dorim Stream.

| Name of the Stream | Area (km²) | Length (km) | Mean Slope of Watershed (%) | Mean Elevation of Watershed (m.a.s.l) | Impervious (%) |
|---|---|---|---|---|---|
| Dorim | 42.50 | 14.51 | 9.99 | 89.34 | 89.85 |

Dorim Stream watershed is an urban watershed suitable for flash flood forecasting to prepare for floods because the stream flows through the city center from its origin to the downstream confluence. The flood warning point of the Dorim Stream is Sindorim Bridge (No. 18), 1500 m upstream of the point of its confluence with the Anyang Stream. It is a candidate high-water mark location for the operation of the envisaged flood forecasting and warning system. The presence of levees makes this location suitable for issuing flood warnings or evacuation orders. Figure 4 shows the cross-section of the stream at the Sindorim Bridge flood warning point.

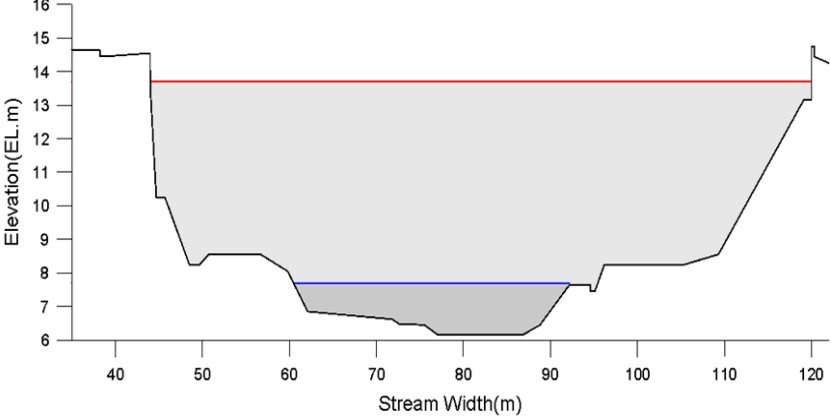

**Figure 4.** Threshold water levels of Dorim Stream at the warning points for issuing a flood warning (blue line) or an evacuation order (red line).

### 3.2. Construction of Rainfall Estimation Model for Flood Warning

It is essential to construct a model that reflects the runoff characteristics as accurately as possible to perform waterfall estimation for flood warning. That is, the model should be constructed in a way that makes the watershed runoff estimation as close as possible to the actual rainfall. To this end, we applied the runoff characteristics of the watershed based on the Stream Maintenance Basic Plan for Anyang Stream Catchment [25], which reflects the current characteristics the most closely among all basic plans published by the local government managing the Dorim Stream and the central government. After applying the parameters determined in the plan and performing correction and calibration, we conducted runoff-dependent water level fluctuation analysis.

For water level analysis according to the flood discharge estimation, we used the cross-section data along the entire stream length of 14.51 km measured at intervals of 50–100 m from the point of its confluence with the Anyang Stream. For accurate water level estimation at the Sindorim Bridge point, which is a prediction point-based on the water level encroachment from the discharge outlet, we set the peak flood discharge based on the design rainfall estimated using the HEC-HMS model as the upstream boundary condition, as shown in Figure 5a. For the hydraulic analysis of the stream channel, we set the known WSs for different backwater conditions of the Anyang Stream as the downstream boundary condition using the HEC-RAS model, as shown in Figure 5b.

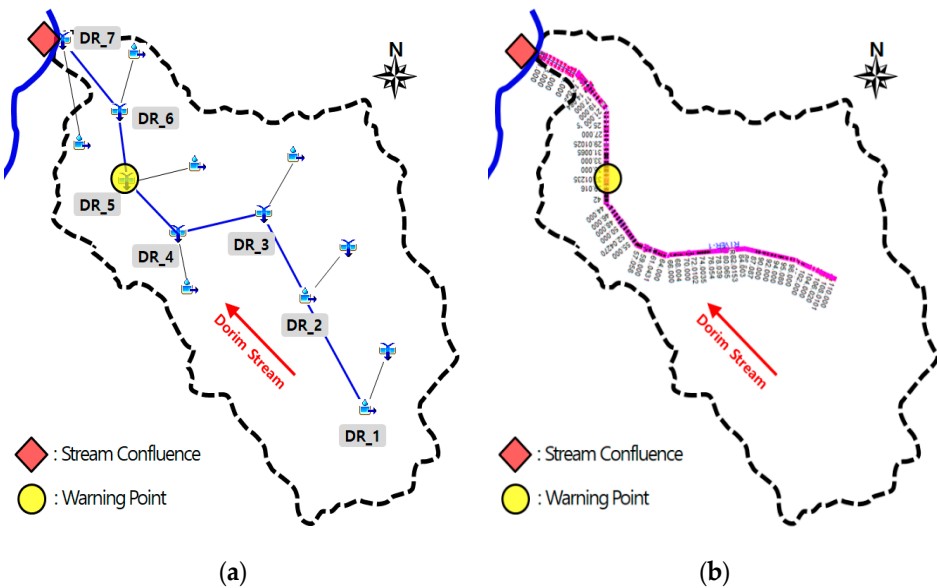

(**a**)　　　　　　　　　　　　　　　　　　(**b**)

**Figure 5.** Rainfall threshold analysis method for flood warning. (**a**) Construction of HEC-HMS model, (**b**) Construction of HEC-RAS model.

This study used a numerical model that includes several methods to simulate the watershed and channel to predict the flow, stage, and timing. Actual data are required to validate the results. This study represents the first step for calibrating and validating the models and ensuring that the models are effective. The second step involves modeling a past flooding event and analyzing the hydraulic response by examining the river discharge and their relationships. The third step involves running the models in the forecast mode using the forecasted rainfall data and comparing the model outputs with the observed information [26]. We constructed a Dorim stream plan based on the river management plan in the Anyang stream area report [25] and conducted the above calibration/validation process. As a result, it was confirmed that the flow rate and the water level are consistent with the design frequency of the stream.

The analysis results revealed that the upstream area covering 5.14 km of the total length of 14.51 km was subjected to the backwater effect due to the effect of the known WS according to the

design frequency at the downstream endpoint, i.e., the point of confluence with the Anyang Stream. The area experiencing the backwater effect stretches to the uppermost end, passing the prediction point at Sindorim Bridge (No. 18). Therefore, because most of the stream sections are expected to undergo abrupt water level fluctuations during flood events, the flood warning time should be estimated by applying the tabulated predicted rainfall amounts presented in this report. Table 3 summarizes the estimated threshold water levels for issuing flood warnings (caution level) and evacuation orders, which correspond to the water levels encroaching into the levee and reaching the embankment, respectively, at the water level prediction point at Sindorim Bridge (No. 18). The threshold water level for flood warning marked by encroachment upon the levee was estimated at m.a.s.l. 7.55 m with a flood discharge rate of 37 $m^3$/s.

**Table 3.** Criteria for issuing a flood warning or an evacuation order for Dorim Stream.

| Warning Criteria | Warning Point (No.) | Minimum Stream Bed Elevation (m.a.s.l) | Threshold Water Level (m.a.s.l) | Threshold Flow Rate ($m^3$/s) |
|---|---|---|---|---|
| Caution | Sindorim Bridge (No. 18) | 6.15 | 7.55 | 37 |
| Evacuation | | | 13.70 | 770 |

*3.3. Estimation of the Rainfall Threshold for Issuing a Flood Warning*

This section describes the process of estimating the value of ΔQ required for the current water level to reach the threshold water levels for issuing a flood warning and an evacuation order at each warning point depending on the water level fluctuations at the downstream points with known WSs. It presents the rainfall rates generating the runoff corresponding to ΔQ along the rainfall duration time series in 30 min intervals (30, 60, 90, and 120 min). Thus, we first checked the current water level (h1), which depends on the rise of the backwater (h2), at the confluence point corresponding to the threshold water levels for issuing a flood warning (Table 4) and an evacuation order (Table 5) at the Sindorim Bridge warning point (No. 18). Then, we employed the results to estimate the additional flow rate (ΔQ) required for the current flow rate (Q1) and water level to reach the threshold water level for flood warning issuance.

**Table 4.** Discharge conditions at point No. 18 for caution warning criteria (unit: $m^3$/s).

| Water Level at No. 0 (h2) | Water Level at No. 18 (h1) | | | | | | | | | | | |
|---|---|---|---|---|---|---|---|---|---|---|---|---|
| | 6.55 m | | 6.75 m | | 6.95 m | | 7.15 m | | 7.35 m | | 7.55 m | |
| | Q1 | ΔQ | Q1 | ΔQ | Q1 | ΔQ | Q1 | ΔQ | Q1 | ΔQ | Q1 | ΔQ |
| 6.35 m | 2 | 33 | 6 | 31 | 11 | 26 | 17 | 20 | 27 | 10 | 37 | 0 |
| 6.45 m | 1 | 32 | 6 | 31 | 11 | 26 | 17 | 20 | 27 | 10 | 37 | 0 |
| 6.65 m | 0 | 0 | 4 | 29 | 9 | 24 | 16 | 19 | 25 | 8 | 36 | 0 |
| 6.75 m | 0 | 0 | 0 | 0 | 8 | 23 | 16 | 19 | 25 | 8 | 36 | 0 |
| 6.85 m | 0 | 0 | 0 | 0 | 6 | 0 | 15 | 18 | 24 | 7 | 35 | 0 |
| 6.95 m | 0 | 0 | 0 | 0 | 0 | 0 | 13 | 16 | 23 | 6 | 35 | 0 |
| 7.05 m | 0 | 0 | 0 | 0 | 0 | 0 | 10 | 0 | 22 | 5 | 34 | 0 |
| 7.35 m | 0 | 0 | 0 | 0 | 0 | 0 | 0 | 0 | 19 | 2 | 32 | 0 |
| 7.55 m | 0 | 0 | 0 | 0 | 0 | 0 | 0 | 0 | 0 | 0 | 0 | 0 |

**Table 5.** Discharge conditions at point No. 18 for evacuation warning criteria (unit: m³/s).

| Water Level at No. 0 (h2) | Water Level at No. 18 (h1) | | | | | | | | | | | |
| --- | --- | --- | --- | --- | --- | --- | --- | --- | --- | --- | --- | --- |
| | 10.00 m | | 11.00 m | | 12.00 m | | 13.00 m | | 13.50 m | | 13.70 m | |
| | Q1 | ΔQ | Q1 | ΔQ | Q1 | ΔQ | Q1 | ΔQ | Q1 | ΔQ | Q1 | ΔQ |
| 9.00 m | 320 | 944 | 573 | 691 | 830 | 434 | 1090 | 174 | 1215 | 49 | 1264 | 0 |
| 9.98 m | 30 | 654 | 503 | 621 | 799 | 403 | 1080 | 164 | 1210 | 49 | 1260 | 0 |
| 10.98 m | 0 | 0 | 44 | 162 | 675 | 130 | 1020 | 104 | 1162 | 47 | 1240 | 0 |
| 11.48 m | 0 | 0 | 0 | 0 | 516 | 94 | 936 | 20 | 1118 | 46 | 1199 | 0 |
| 11.98 m | 0 | 0 | 0 | 0 | 56 | 77 | 803 | 59 | 1036 | 44 | 1124 | 0 |
| 12.48 m | 0 | 0 | 0 | 0 | 0 | 0 | 610 | 54 | 903 | 43 | 1005 | 0 |
| 12.98 m | 0 | 0 | 0 | 0 | 0 | 0 | 66 | 46 | 677 | 42 | 811 | 0 |
| 13.48 m | 0 | 0 | 0 | 0 | 0 | 0 | 0 | 0 | 71 | 41 | 437 | 0 |
| 13.70 m | 0 | 0 | 0 | 0 | 0 | 0 | 0 | 0 | 0 | 0 | 0 | 0 |

Tables 6 and 7 present the rainfall estimates predicted based on the relationship between Q and ΔQ with regard to the presented time series of water levels at Sindorim Bridge. Because the predicted rainfall amount for flood warning issuance at Sindorim Bridge fluctuates depending on the real-time water level observations and the known WS conditions at the confluence point, the typical water level fluctuation results are presented. The higher the known WS relative to the real-time water level observed at the flood warning point, the smaller is the rainfall amount required to reach the threshold water level that may lead to flood warning issuance. In addition, the higher the real-time water level observation relative to the corresponding known WS, the smaller is the predicted rainfall amount for flood warning issuance.

Table 6 shows that when the known WS at the downstream endpoint (the confluence point with Anyang Stream, No. 0) is 6.65 m and the water level observed at the Sindorim Bridge point (No. 18) is 6.75 m, the decision to issue a flood warning is made when the estimated rainfall is 13.2 mm at 30 min, 13.5 mm at 60 min, 13.8 mm at 90 min, and 14.7 mm at 120 min.

**Table 6.** Rainfall depth according to duration for caution warning criteria (unit: mm).

| h2 (min) \ h1 Duration | 6.55 m | | | | 6.75 m | | | | 6.95 m | | | |
| --- | --- | --- | --- | --- | --- | --- | --- | --- | --- | --- | --- | --- |
| | 30 | 60 | 90 | 120 | 30 | 60 | 90 | 120 | 30 | 60 | 90 | 120 |
| 6.35 m | 14.5 | 14.7 | 15.0 | 15.9 | 13.9 | 14.1 | 14.4 | 15.3 | 12.3 | 12.5 | 12.7 | 13.5 |
| 6.45 m | 14.2 | 14.4 | 14.7 | 15.6 | 13.9 | 14.1 | 14.4 | 15.3 | 12.3 | 12.5 | 12.7 | 13.5 |
| 6.65 m | 0.0 | 0.0 | 0.0 | 0.0 | 13.2 | 13.5 | 13.8 | 14.7 | 11.6 | 11.8 | 12.1 | 12.9 |
| 6.75 m | 0.0 | 0.0 | 0.0 | 0.0 | 0.0 | 0.0 | 0.0 | 0.0 | 11.3 | 11.5 | 11.7 | 12.4 |
| 6.85 m | 0.0 | 0.0 | 0.0 | 0.0 | 0.0 | 0.0 | 0.0 | 0.0 | 0.0 | 0.0 | 0.0 | 0.0 |
| 6.95 m | 0.0 | 0.0 | 0.0 | 0.0 | 0.0 | 0.0 | 0.0 | 0.0 | 0.0 | 0.0 | 0.0 | 0.0 |
| 7.05 m | 0.0 | 0.0 | 0.0 | 0.0 | 0.0 | 0.0 | 0.0 | 0.0 | 0.0 | 0.0 | 0.0 | 0.0 |
| 7.35 m | 0.0 | 0.0 | 0.0 | 0.0 | 0.0 | 0.0 | 0.0 | 0.0 | 0.0 | 0.0 | 0.0 | 0.0 |
| 7.55 m | 0.0 | 0.0 | 0.0 | 0.0 | 0.0 | 0.0 | 0.0 | 0.0 | 0.0 | 0.0 | 0.0 | 0.0 |

| h2 (min) \ h1 Duration | 7.15 m | | | | 7.35 m | | | | 7.55 m | | | |
| --- | --- | --- | --- | --- | --- | --- | --- | --- | --- | --- | --- | --- |
| | 30 | 60 | 90 | 120 | 30 | 60 | 90 | 120 | 30 | 60 | 90 | 120 |
| 6.35 m | 10.2 | 10.4 | 10.6 | 11.3 | 6.5 | 6.6 | 6.7 | 7.1 | 0.0 | 0.0 | 0.0 | 0.0 |
| 6.45 m | 10.2 | 10.4 | 10.6 | 11.3 | 6.5 | 6.6 | 6.7 | 7.1 | 0.0 | 0.0 | 0.0 | 0.0 |
| 6.65 m | 9.9 | 10.1 | 10.3 | 10.9 | 5.7 | 5.8 | 5.9 | 6.3 | 0.0 | 0.0 | 0.0 | 0.0 |
| 6.75 m | 9.9 | 10.1 | 10.3 | 10.9 | 5.7 | 5.8 | 5.9 | 6.3 | 0.0 | 0.0 | 0.0 | 0.0 |
| 6.85 m | 9.5 | 9.7 | 9.9 | 10.5 | 5.3 | 5.4 | 5.4 | 5.7 | 0.0 | 0.0 | 0.0 | 0.0 |
| 6.95 m | 8.8 | 8.9 | 9.1 | 9.7 | 4.9 | 5.0 | 5.0 | 5.3 | 0.0 | 0.0 | 0.0 | 0.0 |
| 7.05 m | 0.0 | 0.0 | 0.0 | 0.0 | 4.5 | 4.5 | 4.6 | 4.9 | 0.0 | 0.0 | 0.0 | 0.0 |
| 7.35 m | 0.0 | 0.0 | 0.0 | 0.0 | 3.2 | 3.2 | 3.2 | 3.4 | 0.0 | 0.0 | 0.0 | 0.0 |
| 7.55 m | 0.0 | 0.0 | 0.0 | 0.0 | 0.0 | 0.0 | 0.0 | 0.0 | 0.0 | 0.0 | 0.0 | 0.0 |

**Table 7.** Rainfall depth according to duration for evacuation warning criteria (unit: mm).

| h2 (min) \ h1 Duration | 10.00 m | | | | 11.00 m | | | | 12.00 m | | | |
|---|---|---|---|---|---|---|---|---|---|---|---|---|
| | 30 | 60 | 90 | 120 | 30 | 60 | 90 | 120 | 30 | 60 | 90 | 120 |
| 9.00 m | 132.7 | 138.7 | 146.0 | 155.1 | 102.3 | 106.0 | 111.5 | 118.4 | 70.5 | 72.4 | 75.8 | 80.5 |
| 9.98 m | 97.8 | 101.2 | 106.5 | 113.1 | 93.8 | 96.9 | 102.0 | 108.3 | 66.7 | 68.4 | 71.4 | 75.8 |
| 10.98 m | 0.0 | 0.0 | 0.0 | 0.0 | 37.9 | 38.6 | 39.5 | 41.9 | 33.3 | 33.9 | 34.7 | 36.9 |
| 11.48 m | 0.0 | 0.0 | 0.0 | 0.0 | 0.0 | 0.0 | 0.0 | 0.0 | 27.3 | 27.8 | 28.4 | 30.2 |
| 11.98 m | 0.0 | 0.0 | 0.0 | 0.0 | 0.0 | 0.0 | 0.0 | 0.0 | 24.1 | 24.4 | 25.0 | 26.6 |
| 12.48 m | 0.0 | 0.0 | 0.0 | 0.0 | 0.0 | 0.0 | 0.0 | 0.0 | 0.0 | 0.0 | 0.0 | 0.0 |
| 12.98 m | 0.0 | 0.0 | 0.0 | 0.0 | 0.0 | 0.0 | 0.0 | 0.0 | 0.0 | 0.0 | 0.0 | 0.0 |
| 13.48 m | 0.0 | 0.0 | 0.0 | 0.0 | 0.0 | 0.0 | 0.0 | 0.0 | 0.0 | 0.0 | 0.0 | 0.0 |
| 13.70 m | 0.0 | 0.0 | 0.0 | 0.0 | 0.0 | 0.0 | 0.0 | 0.0 | 0.0 | 0.0 | 0.0 | 0.0 |

| h2 (min) \ h1 Duration | 13.00 m | | | | 13.50 m | | | | 13.70 m | | | |
|---|---|---|---|---|---|---|---|---|---|---|---|---|
| | 30 | 60 | 90 | 120 | 30 | 60 | 90 | 120 | 30 | 60 | 90 | 120 |
| 9.00 m | 39.4 | 40.2 | 41.1 | 43.6 | 17.7 | 17.9 | 18.4 | 19.5 | 0.0 | 0.0 | 0.0 | 0.0 |
| 9.98 m | 38.1 | 38.8 | 39.7 | 42.2 | 17.7 | 17.9 | 18.4 | 19.5 | 0.0 | 0.0 | 0.0 | 0.0 |
| 10.98 m | 29.1 | 29.6 | 30.3 | 32.2 | 17.2 | 17.4 | 17.9 | 19.0 | 0.0 | 0.0 | 0.0 | 0.0 |
| 11.48 m | 9.7 | 9.8 | 10.0 | 10.6 | 17.0 | 17.2 | 17.6 | 18.7 | 0.0 | 0.0 | 0.0 | 0.0 |
| 11.98 m | 20.1 | 20.4 | 20.9 | 22.2 | 16.5 | 16.7 | 17.1 | 18.2 | 0.0 | 0.0 | 0.0 | 0.0 |
| 12.48 m | 19.0 | 19.2 | 19.7 | 20.9 | 16.2 | 16.4 | 16.8 | 17.8 | 0.0 | 0.0 | 0.0 | 0.0 |
| 12.98 m | 17.0 | 17.2 | 17.6 | 18.7 | 15.9 | 16.1 | 16.5 | 17.5 | 0.0 | 0.0 | 0.0 | 0.0 |
| 13.48 m | 0.0 | 0.0 | 0.0 | 0.0 | 15.7 | 15.9 | 16.3 | 17.3 | 0.0 | 0.0 | 0.0 | 0.0 |
| 13.70 m | 0.0 | 0.0 | 0.0 | 0.0 | 0.0 | 0.0 | 0.0 | 0.0 | 0.0 | 0.0 | 0.0 | 0.0 |

## 4. Predicted Rainfall Verification and Validity Analysis

This section describes the process of checking the results presented in Section 3 by comparison with actual water level observations to test the validity of the flash flood warning issuance criteria proposed in this report. In general, urban watershed runoff is characterized by sensitivity to rainfall characteristics. Furthermore, the runoff characteristics that undergo various changes under the influence of the backwater of the main stream at the confluence point affect the selection of the flash flood warning time and warning point. Therefore, the optimal time for flash flood warning issuance should be identified by applying a consistent correlation analysis method throughout the analysis process, from selecting the warning point to obtaining the observed rainfall and water level curves of the given stream from the minimum to the maximum duration. In other words, the identification and estimation of the time for flash flood warning issuance should be determined by considering the efficiency of the decision support related to the flash flood warning issuance customized for all rainfall conditions in line with the purpose of constructing a flood forecasting and warning system.

### 4.1. Identification of Optimal Flash Flood Warning Time

We selected the optimal flash flood warning time by segmenting the water level and rainfall data at regular intervals based on the corresponding real-time observations of the Dorim Stream collected at the Sindorim Bridge point. The collected real-time observations were divided into four segments at intervals of at least 10 min (10, 20, 40, and 60 min), depending on the data characteristics. The maximum interval of 60 min was applied to consider the relatively rapid watershed response to runoff generation according to the actual characteristics of an urban stream.

To test the validity of the flash flood warning time for each duration interval considering the actual water level and rainfall data observed in the watershed, we determined the flash flood warning time using the coefficient of determination ($R^2$). In general, the coefficient of determination is used when analyzing the degree of correlation between two variables ($x$, $y$). In this study, the correlations between two variables associated with various rainfall types were tested based on the observed rainfall curve representing the rainfall during each interval and the corresponding observed water level curve.

Accordingly, $x_n$, and $y_n$ in Equation (1) denote the rainfall and water level, respectively, during each observed interval, and $x$ and $y$ denote their respective mean observed values. The coefficients of determination for the duration intervals were calculated using the standard deviations of each item, $x_i$ and $y_i$, and the number of observed data points $n$:

$$R^2 = \frac{\sum_{i=1}^{n}(x_n - \overline{x})(y_n - \overline{y})}{(n-1)x_i y_i}. \tag{1}$$

As test rainfall events, we used 10 datasets based on the rainfall event observations from July to September 2013 for validation. The rainfall amounts estimated using the proposed flash flood warning method were verified against four rainfall events that exceeded the threshold for issuing a flash flood warning (caution level) out of the 10 real-time observations. Table 8 describes the four rainfall event datasets used for comparison.

**Table 8.** Characteristics of observed rainfall event data (Dorim Stream).

| Event | Starting Time | Ending Time | Rainfall Amount during Investigated Period (mm) |
|-------|---------------|-------------|--------------------------------------------------|
| 1 | 23 July 2013. 07:30 | 23 July 2013. 14:00 | 39.5 |
| 2 | 6 August 2013. 12:40 | 6 August 2013. 15:00 | 49.5 |
| 3 | 23 August 2013. 04:10 | 23 August 2013. 08:00 | 50.0 |
| 4 | 13 September 2013. 00:40 | 13 September 2013. 12:20 | 58.5 |

We calculated the coefficients of determination based on the observed rainfall and water level data by segmenting the duration at intervals between 10 and 60 min to select the duration yielding high values for both sets of coefficients of determination as the optimal flash flood warning time. We chose to use the coefficient of determination because of the wide variation in the standard deviation of the unit rainfall applied for each duration. Given that the behavior diverged from that of the actual observed hydrographs, we considered it necessary to analyze the behavior best reflecting the tendency in the context of efficient operation of flash flood warning. Table 9 presents the calculation results, revealing that the accuracy of the calculated coefficients of determination increases by up to 70% compared with the existing coefficients of determination as the interval increases from 10 min to 60 min. In effect, in contrast to the 26% average reliability with a 10 min interval, over 70% reliability was achieved with the 60 min interval, except for the rainfall event on 8 July 2013, which yielded up to 90.3% reliability, demonstrating that the predicted values well reflect the water level fluctuations. Consequently, we chose 60 min as the optimal interval for flash flood issuance.

**Table 9.** Correlation ($R^2$) between unit rainfall and observed water level.

| Duration / Event | $R^2$ Result (%) | | | |
|------------------|------|------|------|------|
| | 1 | 2 | 3 | 4 |
| 10 min | 46.5 | 20.5 | 15.5 | 34.7 |
| 20 min | 55.6 | 32.1 | 24.5 | 45.1 |
| 40 min | 68.4 | 56.1 | 49.1 | 65.1 |
| 60 min | 69.8 | 78.2 | 74.2 | 76.7 |

*4.2. Comparison with Real-Time Water Level Observations and Improvements*

An urban watershed, which responds sensitively to runoff trends depending on the rainfall characteristics, is affected by the backwater of the mainstream channel, which affects the selection of the flash flood criteria. To estimate the rainfall amount depending on the optimal water level prediction, the discharge outlet opening and closing times should be determined by applying a consistent correlation analysis method throughout the entire process for flash flood warning, from selecting the

warning point to obtaining the observed rainfall and water level curves of the given stream from the minimum to maximum duration.

We selected the optimal prediction time in the regression model using the coefficient of determination ($R^2$) based on the observed water level curve corresponding to the observed rainfall curve. Considering that the water level observation interval of the Dorim Stream is 10 min, we segmented the duration into four intervals (10, 20, 40, and 60 min) for prediction time estimation.

In the estimation results, the highest correlation coefficient was obtained for the longest duration (60 min), which was set as the duration for validity analysis. For the rainfall amount, however, we considered the periods before and after 30 min from the flash flood warning issuance time (t = 0) to ensure safety in consideration of the time gap of 20–30 min between the peak rainfall and peak water level. Thus, a set of three time series (t = −40 ± 20 min, t = −30 ± 30 min, and t = −20 ± 40 min) was considered for warning issuance instead of only one. A flash flood warning is triggered for the relevant watershed when the rainfall amount corresponding to one of these time series exceeds the rainfall threshold estimated in this study.

We tested the validity of the proposed flash flood warning issuance criteria by comparing the predicted values with the corresponding observed values using the datasets for all four rainfall events listed in Table 9 (Tables 10–13).

Out of the 10 major rainfall events that occurred from July to September 2013, we selected four to obtain the validity analysis data based on the observation records for rainfall events exceeding 50 mm/h. Tables 10–13 summarize the application results of each rainfall event.

The real-time observations were used as follows: ① corresponds to the predicted rainfall amount at 60 min; ②, ③, and ④ are the aforementioned set of three time series for flash flood warning issuance depending on the water level prediction considering the safety aspect (① ≤ ② or ③ or ④ leads to flash flood warning issuance); and ⑤ is the time of actual issuance of a flash flood warning for the Dorim Stream, which is used for validity testing by checking the predicted value against it.

In Table 8, the real-time observations used for the validity analysis have the following characteristics: ① corresponds to the 60 min rainfall threshold; and ②, ③, and ④are the aforementioned set of three time series for decision making on flash flood warning issuance considering the safety aspect (① ≤ ② or ③ or ④ meets the conditions for a flash flood warning issuance). In fact, the observed water level reached 6.83 m at 11:40, and the 60 min rainfall threshold for flash flood warning issuance according to the corresponding known WS is 13.26 mm. Although this value is 0.72 m lower than the threshold water level for issuing a caution-level flood warning (7.55 m, as presented in Table 3), the observed water level will reach 7.65 m at 12:40, thus exceeding the 60 min water level threshold. This finding suggests that the flash flood warning should be issued stepwise starting from 11:40 based on the proposed methodology, well ahead of the water level rise to the threshold level. In effect, the rise in water level in the Dorim Stream starting from 11:40 was such that the water level reached the rainfall threshold for flash flood warning issuance pre-calculated for each duration interval at the given point. Moreover, given that observations at 10 min intervals are used stepwise for actual decision making regarding flash flood warning issuance, the methodology proposed in this report is considered suitable for setting the optimal criteria for issuing flash flood warnings and, consequently, capable of supporting decision-making regarding flash flood warning issuance.

Looking at the four heavy rainfall events used for validity analysis, encroachment occurred as per the 60 min rainfall predicted in each of the cases ②, ③, and ④, based on the Dorim Stream levee encroachment water level, 7.55 m. Furthermore, only one of the four heavy rainfall events showed a time difference of 10 min between the predicted and observed levee encroachment times, with the remaining three heavy rainfall events triggering the pumping operation due to encroachment. Finally, based on the comparison of the predicted rainfall amounts with the corresponding observations at 10 min intervals, it can be concluded that stepwise application of the proposed methodology is valid for actual decision making on flash flood warning issuance.

**Table 10.** Comparison of real-time water level observations in Dorim Stream (23 July 2013).

| Date | hh:nn | Observed Water Level | ① Alert Standard Rainfall −60 min- (mm) | ② t = −40 ± 20 min Rainfall (mm) | ③ t = −30 ± 30 min Rainfall (mm) | ④ t = −20 ± 40 min Rainfall (mm) | Alert Check | After 30 min at Water Level | After 40 min at Water Level | After 50 min at Water Level | After 60 min at Water Level | ⑤ Past Record of Pump Operation |
|---|---|---|---|---|---|---|---|---|---|---|---|---|
| ⋮ | ⋮ | ⋮ | ⋮ | ⋮ | ⋮ | ⋮ | ⋮ | ⋮ | ⋮ | ⋮ | ⋮ | |
| 23 July 2013 | 11:20 | 6.79 | 13.57 | 3.0 | 4.0 | 5.5 | | 6.86 | 6.95 | 7.07 | 7.21 | |
| 23 July 2013 | 11:30 | 6.79 | 13.57 | 4.0 | 5.5 | 10.5 | | 6.95 | 7.07 | 7.21 | 7.36 | |
| 23 July 2013 | 11:40 | 6.83 | 13.26 | 5.5 | 10.5 | 18.5 | ○ | 7.07 | 7.21 | 7.36 | 7.65 | |
| 23 July 2013 | 11:50 | 6.86 | 13.02 | 10.5 | 18.5 | 22.5 | ○ | 7.21 | 7.36 | 7.65 | 7.94 | |
| 23 July 2013 | 12:00 | 6.95 | 12.27 | 18.5 | 22.5 | 23.0 | ○ | 7.36 | 7.65 | 7.94 | 8.12 | |
| 23 July 2013 | 12:10 | 7.07 | 11.16 | 22.5 | 23.0 | 26.0 | ○ | 7.65 | 7.94 | 8.12 | 8.25 | |
| 23 July 2013 | 12:20 | 7.21 | 9.39 | 23.0 | 26.0 | 28.0 | ○ | 7.94 | 8.12 | 8.25 | 8.30 | |
| 23 July 2013 | 12:30 | 7.36 | 6.26 | 26.0 | 28.0 | 28.0 | ○ | 8.12 | 8.25 | 8.30 | 8.30 | Start |
| 23 July 2013 | 12:40 | 7.65 | Encroach | 28.0 | 28.0 | 20.5 | ○ | 8.25 | 8.30 | 8.30 | 8.28 | |
| 23 July 2013 | 12:50 | 7.94 | Encroach | 28.0 | 20.5 | 16.5 | ○ | 8.30 | 8.30 | 8.28 | 8.25 | |
| 23 July 2013 | 13:00 | 8.12 | Encroach | 20.5 | 16.5 | 15.5 | ○ | 8.30 | 8.28 | 8.25 | 8.11 | |
| 23 July 2013 | 13:10 | 8.25 | Encroach | 16.5 | 15.5 | 11.5 | ○ | 8.28 | 8.25 | 8.11 | 7.99 | |
| 23 July 2013 | 13:20 | 8.30 | Encroach | 15.5 | 11.5 | 7.5 | ○ | 8.25 | 8.11 | 7.99 | 7.81 | |
| 23 July 2013 | 13:30 | 8.30 | Encroach | 11.5 | 7.5 | 1.5 | ○ | 8.11 | 7.99 | 7.81 | 7.68 | |
| 23 July 2013 | 13:40 | 8.28 | Encroach | 7.5 | 1.5 | 0.5 | ○ | 7.99 | 7.81 | 7.68 | 7.57 | |
| 23 July 2013 | 13:50 | 8.25 | Encroach | 1.5 | 0.5 | 0.5 | ○ | 7.81 | 7.68 | 7.57 | 7.47 | |
| 23 July 2013 | 14:00 | 8.11 | Encroach | 0.5 | 0.5 | 0.0 | ○ | 7.68 | 7.57 | 7.47 | 7.38 | |
| 23 July 2013 | 14:10 | 7.99 | Encroach | 0.5 | 0.0 | 0.0 | ○ | 7.57 | 7.47 | 7.38 | 7.31 | |
| 23 July 2013 | 14:20 | 7.81 | Encroach | 0.0 | 0.0 | 0.0 | | 7.47 | 7.38 | 7.31 | 7.26 | |
| 23 July 2013 | 14:30 | 7.68 | Encroach | 0.0 | 0.0 | 0.0 | | 7.38 | 7.31 | 7.26 | 7.23 | |
| 23 July 2013 | 14:40 | 7.57 | Encroach | 0.0 | 0.0 | 0.0 | | 7.31 | 7.26 | 7.23 | 7.20 | |
| ⋮ | ⋮ | ⋮ | ⋮ | ⋮ | ⋮ | ⋮ | ⋮ | ⋮ | ⋮ | ⋮ | ⋮ | |

**Table 11.** Comparison of real-time water level observations in Dorim Stream (6 August 2013).

| Date | hh:nn | Observed Water Level | ① Alert Standard Rainfall −60 min-(mm) | ② t = −40 ± 20 min Rainfall (mm) | ③ t = −30 ± 30 min Rainfall (mm) | ④ t = −20 ± 40 min Rainfall (mm) | Alert Check | After 30 min at Water Level | After 40 min at Water Level | After 50 min at Water Level | After 60 min at Water Level | ⑤ Past Record of Pump Operation |
|---|---|---|---|---|---|---|---|---|---|---|---|---|
| ⋮ | ⋮ | ⋮ | ⋮ | ⋮ | ⋮ | ⋮ | ⋮ | ⋮ | ⋮ | ⋮ | ⋮ | |
| 6 August 2013 | 12:00 | 6.60 | 14.64 | 0.0 | 0.0 | 0.0 | | 6.60 | 6.60 | 6.60 | 6.60 | |
| 6 August 2013 | 12:10 | 6.60 | 14.64 | 0.0 | 0.0 | 2.0 | | 6.60 | 6.60 | 6.60 | 6.83 | |
| 6 August 2013 | 12:20 | 6.60 | 14.64 | 0.0 | 2.0 | 11.0 | | 6.60 | 6.60 | 6.83 | 6.96 | |
| 6 August 2013 | 12:30 | 6.60 | 14.64 | 2.0 | 11.0 | 18.0 | ○ | 6.60 | 6.83 | 6.96 | 7.10 | |
| 6 August 2013 | 12:50 | 6.60 | 14.64 | 18.0 | 31.5 | 46.0 | ○ | 6.96 | 7.10 | 7.70 | 8.18 | |
| 6 August 2013 | 12:50 | 6.60 | 14.64 | 18.0 | 31.5 | 46.0 | ○ | 6.96 | 7.10 | 7.70 | 8.18 | |
| 6 August 2013 | 13:00 | 6.60 | 14.64 | 31.5 | 46.0 | 49.5 | ○ | 7.10 | 7.70 | 8.18 | 8.52 | |
| 6 August 2013 | 13:10 | 6.83 | 13.26 | 46.0 | 49.5 | 48.5 | ○ | 7.70 | 8.18 | 8.52 | 8.73 | |
| 6 August 2013 | 13:20 | 6.96 | 12.18 | 49.5 | 48.5 | 41.0 | ○ | 8.18 | 8.52 | 8.73 | 8.89 | |
| 6 August 2013 | 13:30 | 7.10 | 10.84 | 48.5 | 41.0 | 34.0 | ○ | 8.52 | 8.73 | 8.89 | 8.91 | |
| 6 August 2013 | 13:40 | 7.70 | Encroach | 41.0 | 34.0 | 20.5 | ○ | 8.73 | 8.89 | 8.91 | 8.87 | Start |
| 6 August 2013 | 13:50 | 8.18 | Encroach | 34.0 | 20.5 | 6.0 | ○ | 8.89 | 8.91 | 8.87 | 8.74 | |
| 6 August 2013 | 14:00 | 8.52 | Encroach | 20.5 | 6.0 | 2.5 | ○ | 8.91 | 8.87 | 8.74 | 8.54 | |
| 6 August 2013 | 14:10 | 8.73 | Encroach | 6.0 | 2.5 | 1.5 | ○ | 8.87 | 8.74 | 8.54 | 8.22 | |
| 6 August 2013 | 14:20 | 8.89 | Encroach | 2.5 | 1.5 | 0.0 | ○ | 8.74 | 8.54 | 8.22 | 7.98 | |
| 6 August 2013 | 14:30 | 8.91 | Encroach | 1.5 | 0.0 | 0.0 | ○ | 8.54 | 8.22 | 7.98 | 7.75 | |
| 6 August 2013 | 14:40 | 8.87 | Encroach | 0.0 | 0.0 | 0.0 | ○ | 8.22 | 7.98 | 7.75 | 7.60 | |
| 6 August 2013 | 14:50 | 8.74 | Encroach | 0.0 | 0.0 | 0.5 | ○ | 7.98 | 7.75 | 7.60 | 7.45 | |
| 6 August 2013 | 15:00 | 8.54 | Encroach | 0.0 | 0.5 | 0.5 | ○ | 7.75 | 7.60 | 7.45 | 7.27 | |
| 6 August 2013 | 15:10 | 8.22 | Encroach | 0.5 | 0.5 | 0.5 | ○ | 7.60 | 7.45 | 7.27 | 7.18 | |
| 6 August 2013 | 15:20 | 7.98 | Encroach | 0.5 | 0.5 | 0.5 | ○ | 7.45 | 7.27 | 7.18 | 7.13 | |
| 6 August 2013 | 15:30 | 7.75 | Encroach | 0.5 | 0.5 | 0.5 | ○ | 7.27 | 7.18 | 7.13 | 7.09 | |
| 6 August 2013 | 15:40 | 7.60 | Encroach | 0.5 | 0.5 | 0.5 | ○ | 7.18 | 7.13 | 7.09 | 7.07 | |
| 6 August 2013 | 15:50 | 7.45 | 3.56 | 0.5 | 0.5 | 0.0 | ○ | 7.13 | 7.09 | 7.07 | 7.03 | |
| 6 August 2013 | 16:00 | 7.27 | 8.33 | 0.5 | 0.0 | 0.0 | ○ | 7.09 | 7.07 | 7.03 | 7.01 | |

**Table 12.** Comparison of real-time water level observations in Dorim Stream (23 August 2013).

| Date | hh:nn | Observed Water Level | ① Alert Standard Rainfall −60 min− (mm) | ② t = -40 ± 20 min Rainfall (mm) | ③ t = -30 ± 30 min Rainfall (mm) | ④ t = -20 ± 40 min Rainfall (mm) | Alert Check | After 30 min at Water Level | After 40 min at Water Level | After 50 min at Water Level | After 60 min at Water Level | ⑤ Past Record of Pump Operation |
|---|---|---|---|---|---|---|---|---|---|---|---|---|
| ⋮ | ⋮ | ⋮ | ⋮ | ⋮ | ⋮ | ⋮ | ⋮ | ⋮ | ⋮ | ⋮ | ⋮ | |
| 23 August 2013 | 3:10 | 6.60 | 14.64 | 0.0 | 0.0 | 0.0 | | 6.60 | 6.60 | 6.60 | 6.60 | |
| 23 August 2013 | 3:20 | 6.60 | 14.64 | 0.0 | 0.0 | 0.0 | | 6.60 | 6.60 | 6.60 | 6.60 | |
| 23 August 2013 | 3:30 | 6.60 | 14.64 | 0.0 | 0.0 | 1.0 | | 6.60 | 6.60 | 6.60 | 6.60 | |
| 23 August 2013 | 3:40 | 6.60 | 14.64 | 0.0 | 1.0 | 5.0 | | 6.60 | 6.60 | 6.60 | 6.77 | |
| 23 August 2013 | 3:50 | 6.60 | 14.64 | 1.0 | 5.0 | 13.0 | | 6.60 | 6.60 | 6.77 | 6.84 | |
| 23 August 2013 | 4:00 | 6.60 | 14.64 | 5.0 | 13.0 | 18.0 | ○ | 6.60 | 6.77 | 6.84 | 6.94 | |
| 23 August 2013 | 4:10 | 6.60 | 14.64 | 13.0 | 18.0 | 24.0 | ○ | 6.77 | 6.84 | 6.94 | 7.02 | |
| 23 August 2013 | 4:20 | 6.60 | 14.64 | 18.0 | 24.0 | 29.0 | ○ | 6.84 | 6.94 | 7.02 | 7.36 | |
| 23 August 2013 | 4:30 | 6.60 | 14.64 | 24.0 | 29.0 | 35.5 | ○ | 6.94 | 7.02 | 7.36 | 7.95 | |
| 23 August 2013 | 4:40 | 6.77 | 13.72 | 29.0 | 35.5 | 38.0 | ○ | 7.02 | 7.36 | 7.95 | 8.31 | |
| 23 August 2013 | 4:50 | 6.84 | 13.18 | 35.5 | 38.0 | 32.0 | ○ | 7.36 | 7.95 | 8.31 | 8.47 | |
| 23 August 2013 | 5:00 | 6.94 | 12.36 | 38.0 | 32.0 | 28.0 | ○ | 7.95 | 8.31 | 8.47 | 8.51 | |
| 23 August 2013 | 5:10 | 7.02 | 11.64 | 32.0 | 28.0 | 22.5 | ○ | 8.31 | 8.47 | 8.51 | 8.51 | |
| 23 August 2013 | 5:20 | 7.36 | 6.26 | 28.0 | 22.5 | 19.0 | ○ | 8.47 | 8.51 | 8.51 | 8.38 | Start |
| 23 August 2013 | 5:30 | 7.95 | Encroach | 22.5 | 19.0 | 12.0 | ○ | 8.51 | 8.51 | 8.38 | 8.20 | |
| 23 August 2013 | 5:40 | 8.31 | Encroach | 19.0 | 12.0 | 6.5 | ○ | 8.51 | 8.38 | 8.20 | 8.02 | |
| 23 August 2013 | 5:50 | 8.47 | Encroach | 12.0 | 6.5 | 5.5 | ○ | 8.38 | 8.20 | 8.02 | 7.82 | |
| 23 August 2013 | 6:00 | 8.51 | Encroach | 6.5 | 5.5 | 5.5 | ○ | 8.20 | 8.02 | 7.82 | 7.65 | |
| 23 August 2013 | 6:10 | 8.51 | Encroach | 5.5 | 5.5 | 6.0 | ○ | 8.02 | 7.82 | 7.65 | 7.48 | |
| 23 August 2013 | 6:20 | 8.38 | Encroach | 5.5 | 6.0 | 5.5 | ○ | 7.82 | 7.65 | 7.48 | 7.34 | |
| 23 August 2013 | 6:30 | 8.20 | Encroach | 6.0 | 5.5 | 5.5 | ○ | 7.65 | 7.48 | 7.34 | 7.25 | |
| 23 August 2013 | 6:40 | 8.02 | Encroach | 5.5 | 5.5 | 5.5 | ○ | 7.48 | 7.34 | 7.25 | 7.19 | |
| 23 August 2013 | 6:50 | 7.82 | Encroach | 5.5 | 5.5 | 4.5 | ○ | 7.34 | 7.25 | 7.19 | 7.18 | |
| 23 August 2013 | 7:00 | 7.65 | Encroach | 5.5 | 4.5 | 4.0 | ○ | 7.25 | 7.19 | 7.18 | 7.16 | |
| 23 August 2013 | 7:10 | 7.48 | 2.53 | 4.5 | 4.0 | 3.0 | ○ | 7.19 | 7.18 | 7.16 | 7.09 | |

**Table 13.** Comparison of real-time water level observations in Dorim Stream (13 September 2013).

| Date | hh:nn | Observed Water Level | ① Alert Standard Rainfall −60 min- (mm) | ② t = -40 ± 20 min Rainfall (mm) | ③ t = -30 ± 30 min Rainfall (mm) | ④ t = -20 ± 40 min Rainfall (mm) | Alert Check | After 30 min at Water Level | After 40 min at Water Level | After 50 min at Water Level | After 60 min at Water Level | ⑤ Past Record of Pump Operation |
|---|---|---|---|---|---|---|---|---|---|---|---|---|
| ⋮ | ⋮ | ⋮ | ⋮ | ⋮ | ⋮ | ⋮ | ⋮ | ⋮ | ⋮ | ⋮ | ⋮ | |
| 13 September 2013 | 2:00 | 6.60 | 14.64 | 4.5 | 5.0 | 5.0 | | 6.83 | 6.85 | 6.85 | 6.84 | |
| 13 September 2013 | 2:10 | 6.72 | 14.05 | 5.0 | 5.0 | 6.5 | | 6.85 | 6.85 | 6.84 | 6.87 | |
| 13 September 2013 | 2:20 | 6.77 | 13.72 | 5.0 | 6.5 | 8.0 | | 6.85 | 6.84 | 6.87 | 6.98 | |
| 13 September 2013 | 2:30 | 6.83 | 13.26 | 6.5 | 8.0 | 7.0 | | 6.84 | 6.87 | 6.98 | 7.21 | |
| 13 September 2013 | 2:40 | 6.85 | 13.10 | 8.0 | 7.0 | 11.5 | | 6.87 | 6.98 | 7.21 | 7.39 | |
| 13 September 2013 | 2:50 | 6.85 | 13.10 | 7.0 | 11.5 | 15.0 | ○ | 6.98 | 7.21 | 7.39 | 7.90 | |
| 13 September 2013 | 3:00 | 6.84 | 13.18 | 11.5 | 15.0 | 16.5 | ○ | 7.21 | 7.39 | 7.90 | 7.99 | |
| 13 September 2013 | 3:10 | 6.87 | 12.94 | 15.0 | 16.5 | 15.5 | ○ | 7.39 | 7.90 | 7.99 | 8.03 | |
| 13 September 2013 | 3:20 | 6.98 | 12.01 | 16.5 | 15.5 | 15.0 | ○ | 7.90 | 7.99 | 8.03 | 8.07 | |
| 13 September 2013 | 3:30 | 7.21 | 9.39 | 15.5 | 15.0 | 14.0 | ○ | 7.99 | 8.03 | 8.07 | 8.07 | |
| 13 September 2013 | 3:40 | 7.39 | 5.43 | 15.0 | 14.0 | 9.5 | ○ | 8.03 | 8.07 | 8.07 | 8.07 | |
| 13 September 2013 | 3:50 | 7.90 | Encroach | 14.0 | 9.5 | 5.0 | ○ | 8.07 | 8.07 | 8.07 | 8.00 | Start |
| 13 September 2013 | 4:00 | 7.99 | Encroach | 9.5 | 5.0 | 3.0 | ○ | 8.07 | 8.07 | 8.00 | 7.90 | |
| 13 September 2013 | 4:10 | 8.03 | Encroach | 5.0 | 3.0 | 3.0 | ○ | 8.07 | 8.00 | 7.90 | 7.76 | |
| 13 September 2013 | 4:20 | 8.07 | Encroach | 3.0 | 3.0 | 2.5 | ○ | 8.00 | 7.90 | 7.76 | 7.62 | |
| 13 September 2013 | 4:30 | 8.07 | Encroach | 3.0 | 2.5 | 2.0 | ○ | 7.90 | 7.76 | 7.62 | 7.48 | |
| 13 September 2013 | 4:40 | 8.07 | Encroach | 2.5 | 2.0 | 1.5 | ○ | 7.76 | 7.62 | 7.48 | 7.36 | |

**Table 13.** *Cont.*

| Date | hh:nn | Observed Water Level | ① Alert Standard Rainfall −60 min- (mm) | ② t = -40 ± 20 min Rainfall (mm) | ③ t = -30 ± 30 min Rainfall (mm) | ④ t = -20 ± 40 min Rainfall (mm) | Alert Check | After 30 min at Water Level | After 40 min at Water Level | After 50 min at Water Level | After 60 min at Water Level | ⑤ Past Record of Pump Operation |
|---|---|---|---|---|---|---|---|---|---|---|---|---|
| 13 September 2013 | 4:50 | 8.00 | Encroach | 2.0 | 1.5 | 1.0 | ○ | 7.62 | 7.48 | 7.36 | 7.27 | |
| 13 September 2013 | 5:00 | 7.90 | Encroach | 1.5 | 1.0 | 1.0 | ○ | 7.48 | 7.36 | 7.27 | 7.16 | |
| 13 September 2013 | 5:10 | 7.76 | Encroach | 1.0 | 1.0 | 0.5 | ○ | 7.36 | 7.27 | 7.16 | 7.09 | |
| 13 September 2013 | 5:20 | 7.62 | Encroach | 1.0 | 0.5 | 0.0 | ○ | 7.27 | 7.16 | 7.09 | 7.02 | |
| 13 September 2013 | 5:30 | 7.48 | 2.53 | 0.5 | 0.0 | 0.0 | | 7.16 | 7.09 | 7.02 | 6.99 | |
| 13 September 2013 | 5:40 | 7.36 | 6.26 | 0.0 | 0.0 | 0.0 | | 7.09 | 7.02 | 6.99 | 6.98 | |
| 13 September 2013 | 5:50 | 7.27 | 8.33 | 0.0 | 0.0 | 0.0 | | 7.02 | 6.99 | 6.98 | 6.96 | |
| 13 September 2013 | 6:00 | 7.16 | 10.11 | 0.0 | 0.0 | 0.0 | | 6.99 | 6.98 | 6.96 | 6.95 | |
| ⋮ | ⋮ | ⋮ | ⋮ | ⋮ | ⋮ | ⋮ | ⋮ | ⋮ | ⋮ | ⋮ | ⋮ | ⋮ |

## 5. Conclusions

To establish an efficient water damage prevention plan for flood damage minimization in urban watersheds, it is essential to prepare an integrated flash flood forecasting and warning system considering the meteorological, hydraulic, and hydrologic characteristics of streams. The implementation of an integrated flash flood forecasting and warning system must be preceded by the development of a system customized to the user demands, which vary depending on the local and disaster characteristics. In view of most flood-induced damage cases occurring in urban watersheds, there is a compelling need to provide appropriate response strategies.

Most studies conducted and systems established in relation to flood forecasting and warning have been focused on large streams and rivers. However, recent years have seen increased flooding events in small streams and related flood damage. In particular, various cultural and leisure spaces created along urban streams are prone to inundation damage, in addition to overflow onto stream embankments. Despite the urgency of countering such flood damage, there are hardly any systematic analysis results regarding urban streams and only a few flash flood forecasting and warning systems are in place.

We established an analysis process for appropriate flash flood forecasting and warning for an urban stream and presented a method for deriving the rainfall thresholds for issuing flash flood warnings. To this end, we developed a rainfall threshold estimation approach enabling short-term prediction based on the real-time stream water level at a flood warning point, as well as the backwater effect of the main stream at the confluence point, and performed analysis to predict the optimal flood warning time depending on the changes in the known downstream WS conditions in the Dorim Stream. We also highlighted the need to consider the effects of water level fluctuations based on the observed upstream and downstream water levels to ensure efficient operation of a flash flood warning system. Based on the results derived, we aim to develop analysis techniques and use them as elementary methods for studying the adequacy of flood forecasting and warning system operation based on reliable validity testing because such techniques are ultimately determined by the criteria for ensuring efficient resident evacuation. The flash flood warning issuance criteria for urban streams proposed in this report are expected to facilitate the issuance of adequate flash flood warnings and setting up of related response strategies considering the duration-dependent rainfall amounts estimated based on scenarios for various backwater conditions. This work is also expected to contribute to the establishment of evacuation-related disaster prevention strategies. However, the short-term rainfall amount estimation proposed in this study is based on the rainfall threshold for estimating the time for flash flood warning issuance for the given stream and is not linked to analysis using radar-based rainfall information and the like. Therefore, it is necessary to develop an integrated model, such as a short-term prediction system, in follow-up research. Furthermore, the most important aspect in flood warning-related research and the selection of warning issuance time is the collection and storage of high-water mark points and high-quality observation data. These elements are still lacking, and continuous observations and validity testing as well as correction of forecasting and warning models are thought necessary. In this regard, the flash flood warning technique and analysis results presented in this report will prove their value only through continuous observation and validation.

**Author Contributions:** Y.S. (Yangho Song) carried out the survey of the literature and wrote the draft of the manuscript; Y.S. (Yangho Song) and Y.P. worked on subsequent drafts of the manuscript; Y.S. (Yangho Song), J.L. and M.P. performed the simulations; and Y.S. (Youngseok Song), J.L. and Y.S. (Yangho Song) conceived the original idea of the proposed method.

**Funding:** This work was funded by the Korea Meteorological Administration Research and Development Program under Grant KMI (2018-03010).

**Conflicts of Interest:** The authors declare no conflicts of interest.

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
