# Peer review of "Flood Forecasting and Warning System Structures: Procedure and Application to a Small Urban Stream in South Korea"

_water, doi:10.3390/w11081571_

Round 1

Reviewer 1 Report

Reviewer’s report

Journal:  Water

Journal’s Ref:  water-534375

Title: Flood Forecasting and Warning System Structures: Procedure and Application to a Small Urban Stream, South Korea

Authors: Song et al. 

Recommendation: Minor Revision

In this paper, the authors presented a framework for flood forecasting and warning system, which could be used for small urban streams. Under this structure, the authors investigate the rainfall threshold for flash flood warming and compared the results with real-time water level observations. They also validated the flash flood warming criteria by comparing the predicted values to observed values. This paper is clear in construction and easy to follow. I recommend the manuscript could be published with minor revision as suggested below:

Comemnt-1: The authors used HEC-HMS and HEC-RAS for the modeling part, but I didn’t see any validation processes in the manuscript, the authors need to point it out in the revised version.

Comment-2: Figure 6-9 are not necessary for this paper, as correlation between rainfall and observed water level have been presented in Table 9, furthermore, Figure 6-9 show the rainfall-water graph not the correlations, the titles for Figure 6-9 are not appropriate. I suggest delete these figures or move them to appendix   

Author Response

We appreciate the reviewer’s helpful comments. We have attempted to reflect the reviewer’s comments in our changes to the manuscript. Detailed descriptions of how these comments were addressed are provided in an attached file. Please note that additions/modification to the original paper is highlighted in red in the revised paper.

Reviewer 2 Report

Dear Authors

You did a wonderful job in this area. However, I have some reservations regarding HEC-RAS modelling coupled with HEC-HMS modelling. From my perspective, HEC-HMS can be replaced with any other suitable semi-distributed hydrological model (e.g. SWAT etc.). Also, I feel that the HEC-RAS section can be improved substantially, for instance, I did not found any manning number for the river morphological study. I did find the cross-sectional analysis section but no clue on how much distance you took this cross-section, kindly describe it in detail. I would like to see an improved and updated version of this important work with some solid recommendations in the current form its just paper without recommendations or outcomes of your work. Sorry, I did not find any novelty in this work.

Author Response

(The authors gave the same response as above.)
